# A tripartite microbial co-culture system for de novo biosynthesis of diverse plant phenylpropanoids

Sierra M. Brooks [1], Celeste Marsan[1], Kevin B. Reed[1], Shuo-Fu Yuan [2], Dustin-Dat Nguyen[2], Adit Trivedi[1], Gokce Altin-Yavuzarslan [3], Nathan Ballinger [4], Alshakim Nelson [3,4] & Hal S. Alper [1,2] ✉

Plant-derived phenylpropanoids, in particular phenylpropenes, have diverse industrial applications ranging from flavors and fragrances to polymers and pharmaceuticals. Heterologous biosynthesis of these products has the potential to address low, seasonally dependent yields hindering ease of widespread manufacturing. However, previous efforts have been hindered by the inherent pathway promiscuity and the microbial toxicity of key pathway intermediates. Here, in this study, we establish the propensity of a tripartite microbial co-culture to overcome these limitations and demonstrate to our knowledge the first reported de novo phenylpropene production from simple sugar starting materials. After initially designing the system to accumulate eugenol, the platform modularity and downstream enzyme promiscuity was leveraged to quickly create avenues for hydroxychavicol and chavicol production. The consortia was found to be compatible with Engineered Living Material production platforms that allow for reusable, cold-chain-independent distributed manufacturing. This work lays the foundation for further deployment of modular microbial approaches to produce plant secondary metabolites.

Plant-derived phenylpropanoids, especially phenylpropene compounds, have distinct flavor and fragrance profiles, possess favorable bioactive properties, and play key roles in plant communication and defense[1–3]. As a result, these compounds are attractive for many industrial applications and are often harvested from non-traditional crops which presents challenges to supply chain availability and purity. For example, eugenol is typically isolated from clove oil and has varied applications including as an antifungal and antimicrobial agent, as a pharmaceutical agent for antioxidant, anti-inflammatory, and anti-cancer traits as well as uses as a dental anesthetic[4–6]. Not surprisingly, there is high demand for eugenol with a current estimated market size of over 550 million U.S. dollars ($)[7]. Clove essential oil (containing 80% eugenol) can be extracted from clove buds at a yield of 10–20%[8,9]. The

highest density clove farms yield approximately 22 kilogram (kg) clove oil per acre[9,10], corresponding to ~17.6 kg eugenol per acre of clove trees. These efforts are complicated by the fact that (like most natural plant products) eugenol levels vary by season[6], clove plants require long lead times (~1–20 years) to reach harvest maturity, and clove bud harvesting requires extensive manual labor to not disrupt growth and future clove yields[11].

Beyond eugenol, other allylbenzenes such as hydroxychavicol and chavicol are likewise of industrial interest as flavors and fragrances, antifungals, and cancer therapeutics[12–14], as well as precursors for antimicrobial polymers for applications as dental cements[15], high-performance thermosets[16], anti-inflammatory therapeutics[17], and materials for environmental remediation[18]. Hydroxychavicol can be

[1]McKetta Department of Chemical Engineering, The University of Texas at Austin, Austin, TX, USA. [2]Institute for Cellular and Molecular Biology, The University of Texas at Austin, Austin, TX, USA. [3]Molecular Engineering and Sciences Institute, University of Washington, Seattle, WA 98195, USA. [4]Department of Chemistry, University of Washington, Box 351700 Seattle, WA, USA. ✉e-mail: halper@che.utexas.edu

extracted from the Piper betel plant, but extraction rates are highly variable with poor yields[19]. Chavicol accumulation in plants is quite rare[2], with its counterpart methyl chavicol being the main component of tarragon essential oil[20].

Alternative methods to plant-based extraction of these natural products would greatly improve access while also reducing supply chain uncertainty and cost. To this end, cell-based biosynthesis using the approaches of metabolic engineering, synthetic biology, and genomics, has emerged as a promising alternative to both organic synthesis and plant extraction. This approach is supported by many recent examples of microbially produced plant secondary metabolites including breviscapine[21], opioids[22], cannabinoids[23], dencichine[24], tropane alkaloids[25], and vinblastine[26]. Using similar approaches, we posit that microbial production of phenylpropenes can offer a sustainable alternative to current plant-based production methods while also supplementing a challenging supply chain required to meet growing global demand. Supporting this premise, efforts have demonstrated phenylpropene production from monolignol and hydroxycinnamic acids[14,27], but de novo syntheses from simple sugars have not been realized.

A high number of heterologous enzymatic steps, pathway promiscuity[3,28], and microbial toxicity of pathway intermediates[4,13,29] hinder efforts to effectively engineer a singular microbial strain from robust phenylpropenes synthesis. For clarity, this text defines promiscuity within the phenylpropene pathway as the ability of multiple pathway enzymes to perform the same type of reaction on multiple substrates (referred to as substrate promiscuity in other contexts[30]). For instance, the set of enzymes responsible for conversion of ferulic acid to coniferyl alcohol has also been shown to readily act on available pools of coumaric acid and caffeic acid in prior works[28]. To address similar challenges for other products, recent efforts have demonstrated the value of co-culture engineering strategies as a means of reducing metabolic burden of long synthetic pathways, minimizing buildup of toxic intermediates, and maximizing system modularity[28,31,32]. Furthermore, this strategy could potentially bypass known promiscuity of enzymes in the phenylpropene pathway[14,28] by splitting pathways prior to a promiscuous step. Likewise, co-culture strategies can enable plug and play swapping of upstream enzymes to allow for de novo production of multiple, distinct phenylpropenes.

One of the complications of microbial co-culturing is supporting the compatibility and stability of the individual organisms in the consortia. These aspects can make co-culturing challenging and less-portable than mono-culturing approaches. To address this issue, bioproduction schemes using Engineered Living Materials (ELMs) have emerged as a viable strategy to yield reusable biocatalysts and effectively control consortium dynamics for production of compounds ranging from small molecules to large proteins[31,33]. These ELM-based approaches can uniquely support stable co-cultures by spatially encapsulating individual microbes to avoid growth-based competitions that can occur during outgrowth and repeated culturing. Additionally, ELM bioproduction can enable on-demand production without cold-chain requirements, with over one year of shelf stability for encapsulated fungal bioproduction systems[34], thus enabling resource-limited biomanufacturing[35]. On-demand phenylpropene production would be advantageous as these compounds have limited stability at elevated temperatures[36,37].

In this study, we designed a three-member microbial co-culture to enable de novo production of phenylpropenes. We initially establish a eugenol production scheme and then show that downstream enzyme promiscuity and modular design can be leveraged to achieve the production of hydroxychavicol and chavicol from simple sugar feed sources. Finally, we demonstrate the system's compatibility with ELM-based production schemes as a method to increase biocatalytic stability, reusability, and potential in distributed manufacturing schemes. This work highlights the power of leveraging co-culture design and enzyme promiscuity to yield sustainable production for an array of industrially attractive plant secondary metabolites.

## Results

### Design and construction of a tripartite co-culture system for phenylpropene biosynthesis

The phenylpropene biosynthetic pathway was divided into three modules each developed in *Escherichia coli* (*E. coli*), with eugenol as the initial target: Module I, p-Coumaric acid biosynthesis; Module II, Ferulic acid biosynthesis; and Module III, Eugenol biosynthesis (Fig. 1). A collection of known and herein tested modifications were selected for each of these modules. For Module I, high p-coumaric acid production from glycerol and glucose can be achieved via knockout of the TyrR regulator[38] and heterologous overexpression of the highly active and specific tyrosine ammonia lyase from *Flavobacterium johnsoniae*[39]. For Module II, codon-optimized versions of enzymes C3H from *S. espanaensis* and COMT from *Arabidopsis thaliana* were selected as they have previously resulted in the highest reported production of ferulic acid from tyrosine[40]. For Module III, the final biosynthetic pathway for eugenol was constructed via previously characterized enzymes for intermediates. The first three biosynthetic enzymes included 4CL1 from *A. thaliana*, CCR from *Leucaena leucocephala*, and ADH6 from *Saccharomyces cerevisiae* based on previous reports of generating coniferyl alcohol in *E. coli*[28]. For the final two steps, CFAT and EGS from *P. hybrida* were selected as these enzymes have been characterized in *E. coli*[41].

The underlying premise of this co-culture approach is the necessity to minimize side product buildup and metabolic burden on an individual strain[28,32,40]. This separation is also especially poignant given the known promiscuity of enzymes in this phenylpropanoid pathway as described above. Prior reports engineering this pathway have shown a quick buildup of the toxic intermediate coumaric acid[40], on which Module III enzymes are known to act[28]. To highlight the necessity of this modular, co-culture strategy for these products, we also incorporated all three modules (Modules I-III) into a singular *E. coli* strain (Fig. 2A). As expected, the combined strain failed to produce any detectible levels of eugenol (Fig. 2C). A side product detected in this experiment with a retention time shortly after the eugenol standard was confirmed by LC-MS to not contain any of the target phenylpropenes of this work (eugenol, chavicol, or hydroxychavicol). Moreover, its appearance in the coumaric acid mono-culture suggests it is likely a metabolite related to upstream tyrosine or coumaric acid biosynthesis (Supplementary Fig. 1). To bypass these issues, the tripartite co-culture approach (Fig. 2B) was evaluated by using a 1:3:1 ratio of strains (presuming a bias needed for Module II to boost ferulic acid production as later validated in Fig. 3D and Fig. 4C). Using this approach, de novo eugenol production was observed (Fig. 2C). Having demonstrated efficacy of this tripartite co-culture approach for de novo eugenol production, efforts to increase titer were explored.

### Maximizing de novo ferulic acid titer for improved downstream eugenol titer

Initial optimization efforts focused on improving Modules I and II (Fig. 3A) to maximize ferulic acid production with minimal coumaric acid buildup using a media optimization strategy. Generation of a BL21(DE3) strain with Module I yielded $292 \pm 5$ mg/L coumaric acid in 48 h (Fig. 3B) in a base modified MOPS media. Prior studies on this pathway vary drastically in media composition[40,42]. As a result, ingredients in modified MOPS media were systematically added and removed from the base recipe to determine individual components' significance. The addition of iron, previously shown to boost caffeic acid production via action as a redox partner for coumarate-3-hydroxlases[39], and yeast extract, previously shown to boost cell health and phenylpropanoid pathway compounds production[28], aided in boosting ferulic acid production with slightly decreased coumaric

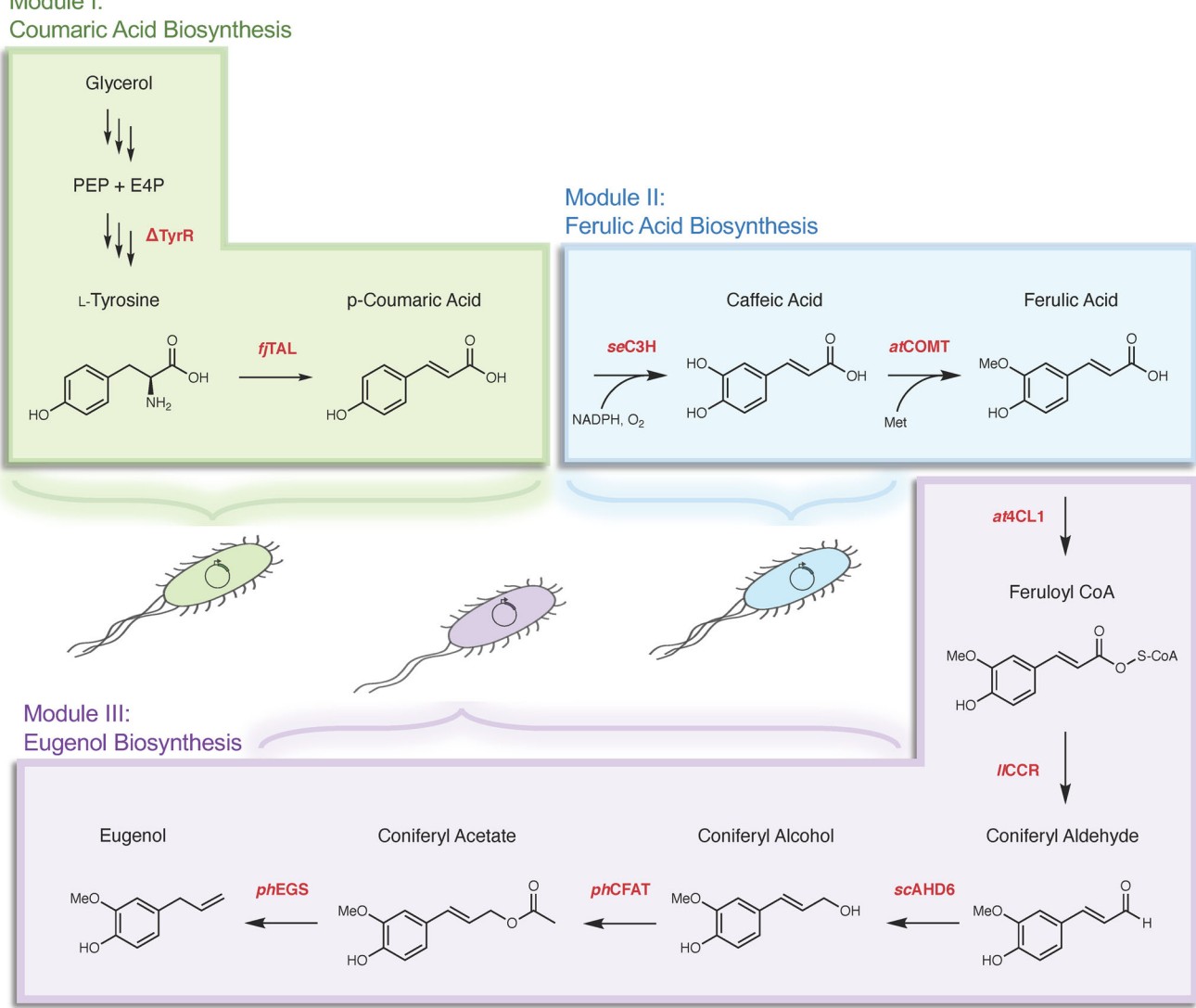

**Fig. 1 | Overview of biosynthetic pathway and tripartite modular scheme for phenylpropene synthesis.** Module I (coumaric acid biosynthesis module) includes modifications to boost intracellular tyrosine flux, as well as heterologous expression for efficient conversion of L-tyrosine to p-coumaric acid. Module II (ferulic acid biosynthesis module) incorporates heterologous enzymes with high efficiency for conversion of hydroxycinnamic acid substrates to ferulic acid in bacteria. Module III (eugenol biosynthesis module) includes heterologous enzymes for conversion of hydroxycinnamic acid substrates to phenylpropenes with high efficiency. Heterologous enzymes: *fj*TAL, tyrosine ammonia lyase from *F. johnsoniae*; *se*C3H, p-coumarate-3-hydroxylase from *S. espanaensis*; *at*COMT, caffeic acid-o-methyltransferase from *A. thaliana*; *at*4CL1, 4-coumarate-CoA ligase from *A. thaliana*; *ll*CCR, cinnamoyl-CoA reductase from *L. leucocephala*; *sc*ADH6, alcohol dehydrogenase 6 from *S. cerevisiae*; *ph*CFAT, coniferyl alcohol acyltransferase from *P. hybrida*; *ph*EGS, eugenol synthase from *P. hybrida*.

acid buildup (Fig. 3C). In an effort to remove the complexity of yeast extract in media, we first identified that the vitamins pantothenic acid, niacin, and riboflavin could enhance production in minimal media while also lowering the prevalence of coumaric acid buildup. We identified a final formulation supplemented with iron, pantothenic acid, niacin, riboflavin, and gluconate that removes the need for yeast extract supplementation while also supporting the highest ferulic acid production across all formulations tested (Fig. 3C). These improvements can be rationalized as pantothenic acid and niacin play key roles in boosting cellular metabolism[43,44], whereas riboflavin and gluconate have been shown to boost iron availability[45] and NADPH regeneration[46], respectively. Additionally, pantothenic acid leads to production of Coenzyme A[43], which bolsters cell health and likely would act synergistically in the 4CL step used in Module III.

Using this final formulation, a co-culture system comprised of Module I and II was evaluated. This scheme enabled the highest ferulic acid production, with minimal coumaric acid buildup, at a Module I:

Module II ratio of 1:3 (Fig. 3D). This co-culture achieved a production titer of $363.1 \pm 28.9$ mg/L ($1.87 \pm 0.15$ mM) ferulic acid in 2 ml culture volume from a combined glycerol and glucose feed, representing the highest reported ferulic acid titer to date (Fig. 3E). Notably, since no yeast extract was utilized in this final optimized media formulation, production is truly de novo from the fed glycerol and glucose in this fully defined, minimal media condition.

**Pathway optimization for fully de novo eugenol production**
After improving metabolic flux through the first two modules, the capacity of Module III to efficiently convert pathway intermediates into eugenol was assessed. Module III was divided into two cassettes: one with the first three steps previously demonstrated in *E. coli*[28] and the other with the Petunia genes for conversion of coniferyl acetate to eugenol (Fig. 1). A set of feeding assays, in which cells were fed 2 mM substrate, demonstrated that both sub-pathways in this module capable could effectively convert ferulic acid to coniferyl alcohol (SMB151)

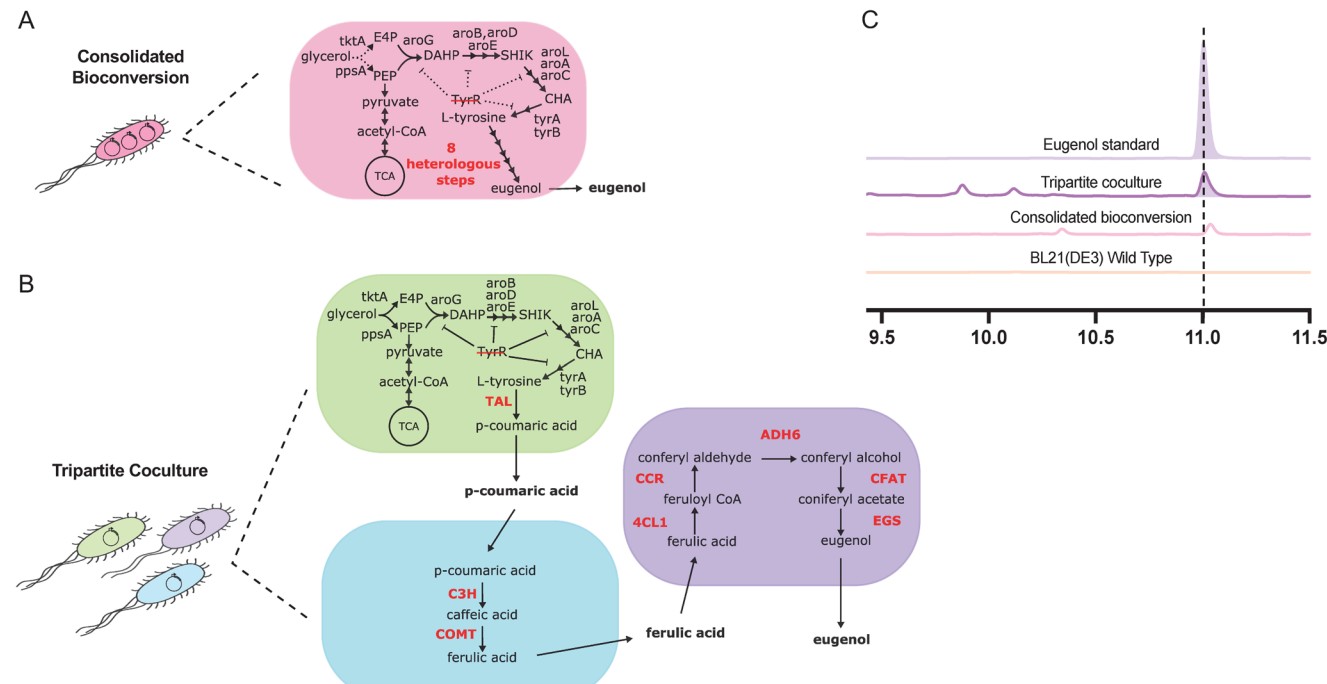

**Fig. 2 | Tripartite co-culture design enables de novo eugenol biosynthesis. A** Consolidated bioconversion strain design containing endogenous tyrosine pathway modifications and 8 heterologous enzymes for de novo eugenol production in a single strain. **B** Tripartite co-culture design in which Modules I-III were separated into three distinct strains to minimize toxic side product buildup and appropriately distribute metabolic burden. **C** Representative high-performance liquid chromatography (HPLC) analysis results comparing consolidated bio-conversion (unsuccessful) and tripartite co-culture (successful) strategies for eugenol production, with comparison to an analytical standard, as well as negative control (wild type BL21(DE3) *E. coli*). The expected product location is represented by a vertical dashed line.

and coniferyl alcohol to eugenol (SMB146), respectively. Likewise, the combined Module III strain (SMB150) was suitable for converting ferulic acid feed to eugenol with minimal buildup of the coniferyl alcohol intermediate (Fig. 4A). With functionality of this Module III validated, a design space of co-culture ratios was evaluated using the Module I: Module II ratio of 1:3 determined earlier (Fig. 3D). Assessment of this design space yielded an optimum Module I: Module II: Module III ratio of 1:3:1, with regard to maximizing eugenol titer after 63 h (Fig. 4B). This co-culture ratio led to production of $50.6 \pm 0.6$ mg/L eugenol in a 1 ml deep well plate (Fig. 4C) and a maximum titer of $66 + 7.5$ mg/L ($0.40 \pm 0.046$ mM) achieved in 30 ml shake flasks upon initial scale-up (Fig. 4D). This conversion (nearly 22% of stoichiometric maximum ferulic acid titer on a molar basis) utilized a 4x lower cell density as the ferulic acid consortia (as detailed in the methods section), making the conversion on a per-cell basis comparable to that of the feeding assays.

**Accessing de novo hydroxychavicol and chavicol biosynthesis via modular construction**

Using the optimized pathways above, we sought to demonstrate the potential to quickly enable production of other phenylpropenes, specifically hydroxychavicol and chavicol, by exploiting the modularity of the consortia-based approach and known promiscuity of the Module III enzymes for additional hydroxycinnamic acid substrates. In particular, the biosynthetic routes for these compounds leverage many of the same pathway enzymes, with hydroxychavicol and chavicol production possible through activity of Module III on caffeic acid and coumaric acid substrates, respectively (Fig. 5A–C). To achieve de novo hydroxychavicol production, we modified Module II via removal of the COMT enzyme (Fig. 5B). In a similar vein, we designed a chavicol-producing consortium via removal of Module II altogether (Fig. 5C), which was facilitated via our use of Module II as an individual strain in

the initial design. By employing the earlier optimized modular ratio of 1:3:1 to the COMT-modified Module II, we enabled formation of hydroxychavicol as the major phenylpropene product (Fig. 6A, C). Of note, hydroxychavicol was also formed as a product in the full eugenol consortia (Fig. 6A, B), due to the activity of Module III enzymes on caffeic acid before full conversion by the methyltransferase. The absolute titers of hydroxychavicol observed within the complete tripartite co-culture and the tripartite co-culture [-COMT] were comparable (Supplementary Table 4). These results (specifically the tripartite co-culture [-COMT] system) allow for the direct production of hydroxychavicol without the need to separate hydroxychavicol from eugenol.

For chavicol production, we utilized a ratio of Module I: Module III (1:1), with complete removal of Module II (Fig. 5C), which resulted in accumulation of chavicol as the sole phenylpropene product (Fig. 6A, C). Thus, the coupled effects of enzyme promiscuity and modular pathway separation enabled facile control over phenylpropene product distribution. Given the unreliable production levels, as well as variable seasonal yields, of eugenol and hydroxychavicol in plants, this co-culture platform could provide a facile method to produce desired phenylpropenes with control over production level and yields. As chavicol is not typically detected in plants, this system could provide an option for expanded production and distribution of this product. Overall, these results highlight the potential to use the same consortia platform backbone to alter the major phenylpropene product formed in a given fermentation based on oscillating market demands via plug and play swapping of upstream enzymes.

**Engineered living material-based production extends biocatalyst efficiency and reusability of this consortia**

To bypass the challenges of liquid culture-based production flexibility and co-culture stability, the use of ELMs has been

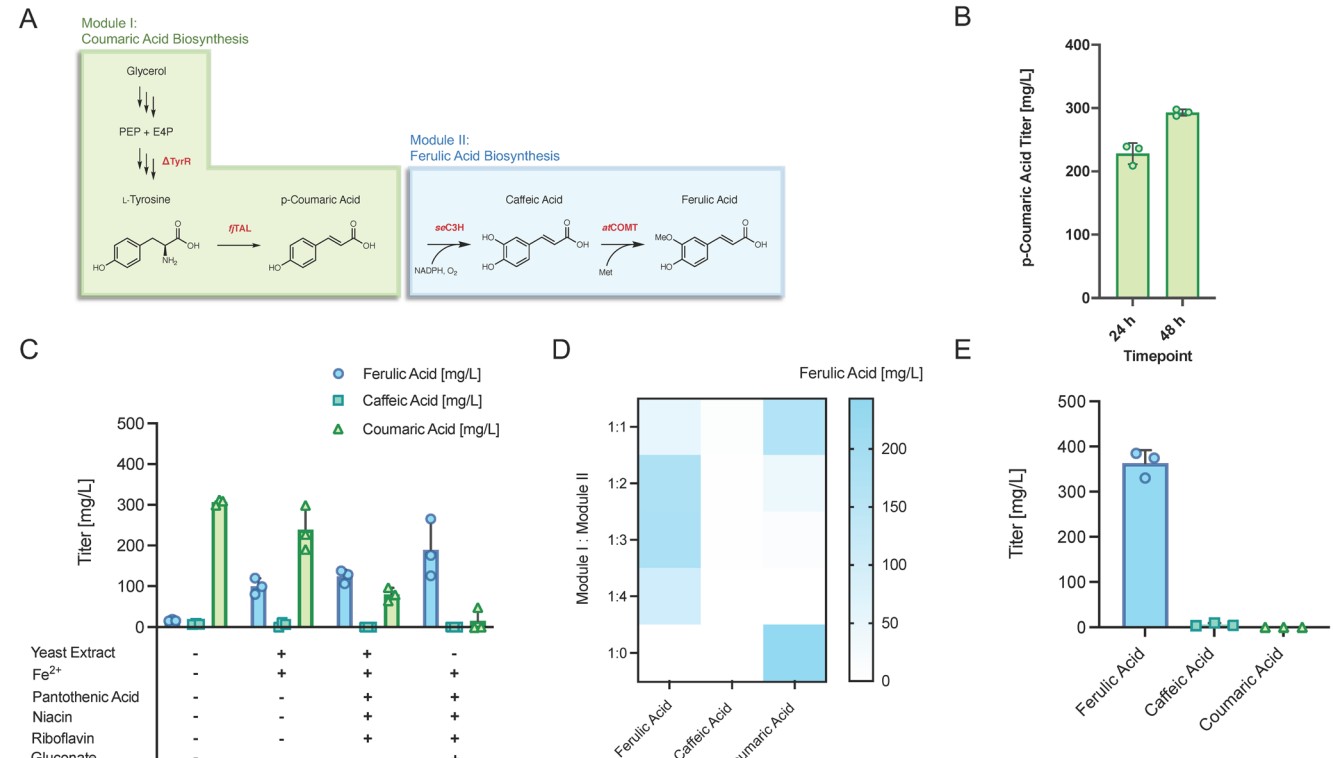

**Fig. 3 | Production of highest de novo ferulic acid titer via modular strain and media optimization. A** Modular scheme for bioproduction of ferulic acid from a glycerol feed, utilizing three of the heterologous enzymes detailed into Fig. 1. **B** p-coumaric acid titer achieved by Module I in modified MOPS media after 24 and 48 h culture in 2 ml tubes. **C** Titers of p-coumaric acid, caffeic acid, and ferulic acid after 63 h in modified MOPS media by Modules I and II resulting from systematic addition or subtraction of additional media components. **D** Heat map of p-coumaric acid, caffeic acid, and ferulic acid titers achieved via optimized media formulation across a range of Module I: Module II ratios. **E** 63 h titers achieved by optimized module and media conditions at 2 ml tube scale, demonstrating highest de novo ferulic acid production. Data are mean ± SD.; $n = 3$ biological replicates. Source data are provided as a Source Data file.

proposed[31,33,34]. Here, we sought to apply the principles of encapsulation-based production to 2- and 3- part consortia for ferulic acid and eugenol production, respectively. Both of these consortia were verified to be unstable in liquid repeated-batch culture mode and failed to produce the target compounds following a single round of fermentation (Fig. 7A, C). This lack of culture stability is caused by significant change in strain ratios observed between the beginning and end of the production stage (Supplementary Table 3). ELM-based production was compared directly with liquid culture-based production for both separate and combined outgrowth culture modes to determine efficiency and robustness in different bioprocessing formats. In brief, these two modes entailed either (1): outgrowth of each module in separate vessels followed by combination of modules into the same vessel at the intended co-culture ratio for the production stage, or (2): outgrowth of each module in the desired co-culture ratio together in the same vessel followed by direct transfer to the production stage.

Ferulic acid production from ELM-based cultures was evaluated for optimal co-culture ratios enabled by addition of varied amounts of equally sized ELMs (e.g. a 1:2 ratio indicates one, 50 mg ELM containing Module I and two, 50 mg gels containing Module II, all created with a constant cell inoculum density). In this scheme, separate outgrowths followed by combination of gels was compared to the case where the gels' outgrowth was allowed to happen together (comparing top schematics of Fig. 7A, B). ELM-based ferulic acid production by the co-culture constructed via separate outgrowth demonstrated fairly consistent ferulic acid production across tested co-culture ratios, with the highest selectively for ferulic acid over coumaric or caffeic acid observed at the 3:1 ratio (Fig. 7A), consistent with findings for the liquid cultures (Fig. 3D). In the case of combined outgrowth and production,

the highest production and specificity towards ferulic acid was also observed for the 1:3 ratio (Fig. 7B). Interestingly, the accumulation of precursors coumaric acid and caffeic acid, as well as final ferulic acid titers, differed greatly between separate and combined ELM culture modes (Fig. 7A, B). The exact mechanism underlying these phenomena is unclear. To make a more accurate comparison of per-cell production in liquid vs ELM cultures, a normalized production of titer (mg/L) per total saturated culture loading (ml) into the production media was used. For both combined and separation outgrowth/production modes, the ELM cultures outperformed the liquids in terms of biocatalytic efficiency and especially repeatability (Fig. 7A, B). Specifically, the ferulic acid ELMs were able to be reused by simply transferring these elements to fresh media, whereas the liquid cultures were limited in catalytic activity to a single batch use. This phenomenon has been seen previously in ELM vs liquid cultures[31], and is likely due to the ability of the polymeric scaffold to prevent cell-growth competition seen in liquid cultures with cells possessing differential growth rates or other advantages.

In the case of eugenol, production was only observed in both liquid and ELM-based production schemes when modules were grown out separately (Fig. 7C, D), with the optimum ratio again agreeing with that of the liquid culture (1:3:1). The need for this separate outgrowth stage is likely due to the consortia instability and depletion of Module I during stages in which all modules are grown together (Supplementary Table 3). This instability could arise from a variety of mechanisms including intermediate product toxicities[4,13,29] or a growth advantage of Module II, potentially leading to depletion of Module I over time. Further investigation into this phenomenon is warranted in future studies to elucidate the exact impact of combined vs separate outgrowth on ELM production. In the case of separate outgrowth and

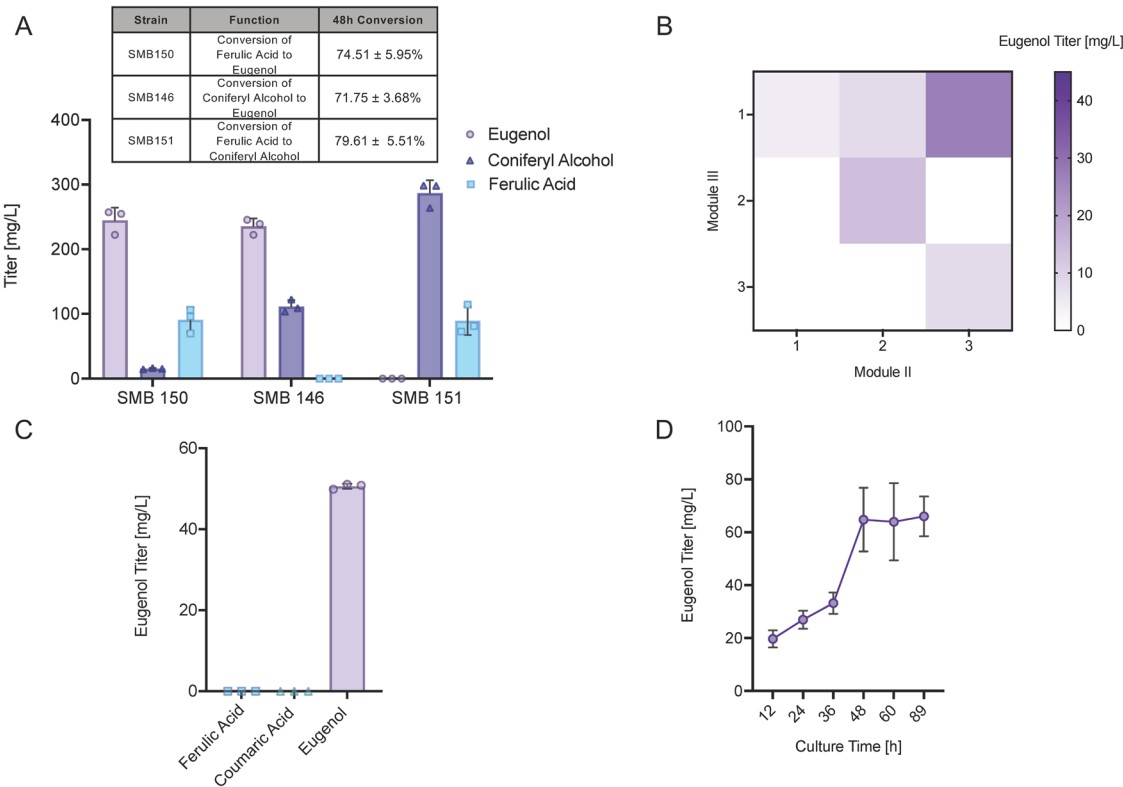

**Fig. 4 | Pathway optimization maximizes de novo eugenol production.**
**A** Production of eugenol and key intermediates via full Module III (SMB150) as well as ferulic acid to coniferyl alcohol (SMB151) and coniferyl alcohol to eugenol (SMB146) portions. Table containing molar conversion of 2 mM substrates after 48 h is included above graph. **B** Heat map of eugenol titer achieved across a range of Module II: Module III ratios (Module I ratio held at 1). **C** Eugenol titer achieved by optimized tripartite co-culture at 1 ml deep well scale, demonstrating minimal accumulation of hydroxycinnamic acid side products. **D** Eugenol titer of tripartite co-culture at 30 ml flask scale, demonstrating first and highest de novo phenylpropene bioproduction. Data are mean ± SD.; n = 3 biological replicates. Source data are provided as a Source Data file.

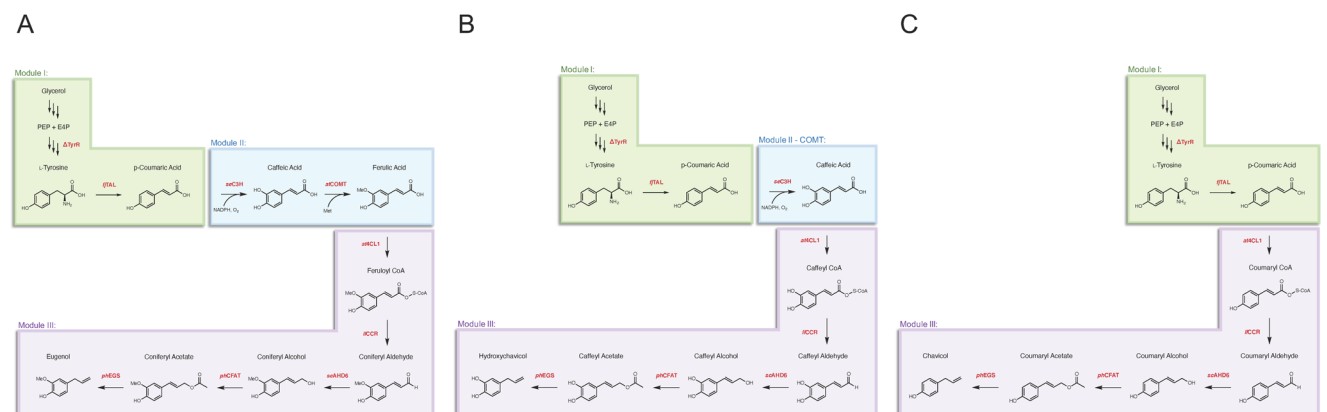

**Fig. 5 | Employing pathway modularity to diversify phenylpropene biosynthesis.** Overview of modular arrangements to synthesize distinct phenylpropene products via downstream enzyme promiscuity for diverse hydroxycinnamic acid pathway intermediates. **A** Use of complete forms of Modules I-III to produce eugenol. **B** Removal of COMT from Module II to direct flux to hydroxychavicol production. **C** Removal of Module II entirely to enable chavicol accumulation.

production, liquid-based cultures showed an initial higher eugenol titer per-cell loading. However, only ELM-based cultures enabled a reuse of the culture in subsequent batch processing (Fig. 7C), thus demonstrating the importance of this approach for complex consortia. Overall, ELM-based culturing was shown to be suitable for both ferulic acid and eugenol, with successful module ratios matching those observed in liquid culture modes, as well as expanded module ratio options for the case of ferulic acid. While only a single round of reuse was tested in this work as a proof-of-concept, we anticipate that multiple additional rounds of reusability could be attainable as shown in prior works showcasing reusability of F127-BUM ELMs[31,33]. These results demonstrate a potential flexibility of ELM-based production of phenylpropene in a manner that can enable cold-chain independent production from a stabilized tripartite consortium. In particular, previously described preservability of ELM-based production enables a stable co-culture to be transported or used on-demand.

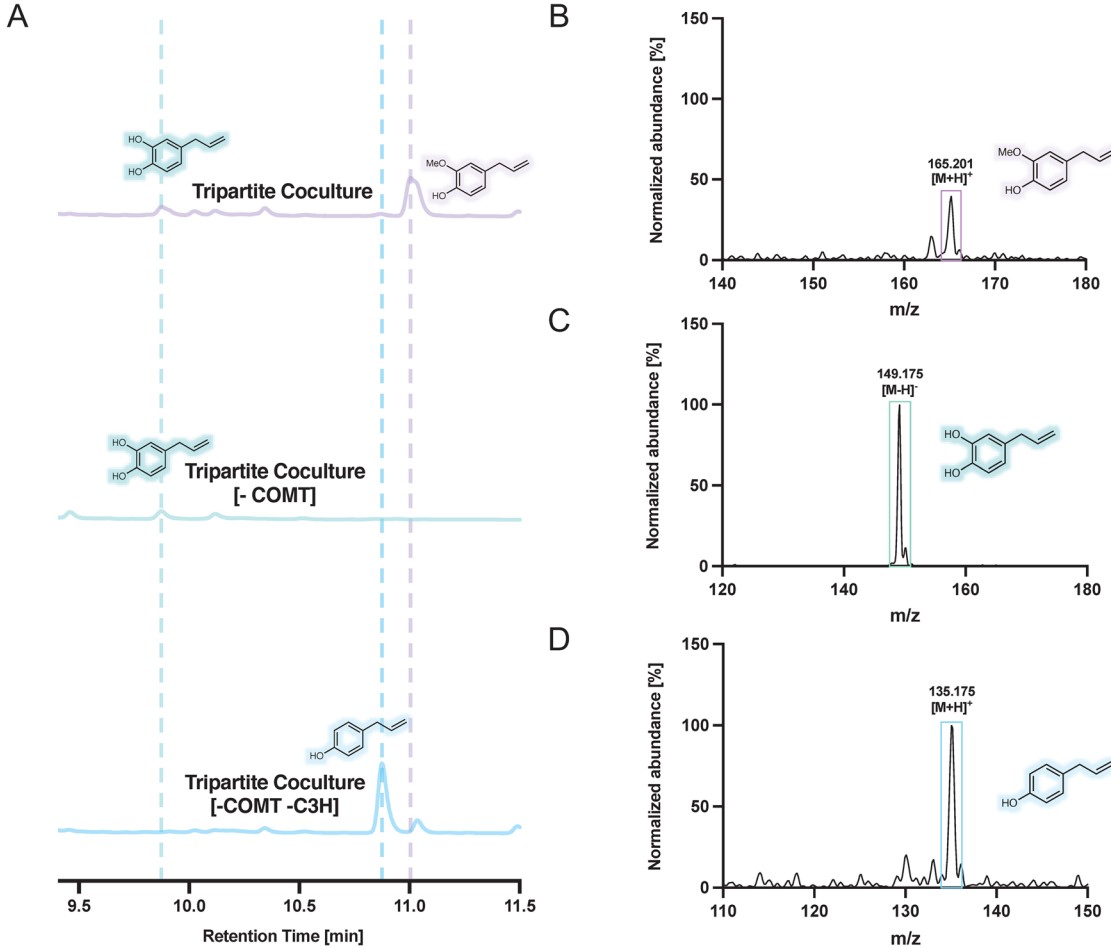

**Fig. 6 | Verification of diverse phenylpropene biosynthesis achieved via modular rearrangement. A** High-performance liquid chromatography (HPLC) analysis of products from the modular schemes in Fig. 5 (top: Fig. 5A, middle: Fig. 5B, bottom: Fig. 5C). Hydroxychavicol, chavicol, and eugenol retention times indicated via green, blue, and purple lines, respectively. **B** LC−MS results of the tripartite co-culture verifying eugenol production in positive mode. $m/z$ value is for the parent ion $[M + H]^+$. **C** LC−MS results of the tripartite co-culture [-COMT] verifying hydroxychavicol production in negative mode. $m/z$ value is for the parent ion $[M − H]^-$. **D** LC−MS results of the tripartite co-culture [-COMT -C3H] verifying chavicol production in positive mode. $m/z$ value is for the parent ion $[M + H]^+$.

## Discussion

Biosynthetic production of phenylpropenes presents a sustainable alternative to plant-based production methods, which are hampered by low yields, seasonal variation, reliance on strenuous manual labor, and geographical restrictions. Herein, we demonstrate that the high metabolic burden and enzyme promiscuity associated with this pathway can be bypassed through a tripartite microbial co-culture system. Furthermore, the interfacing of engineered microbial systems with engineered polymeric matrices has enabled ELM-based bioproduction for small molecule and protein targets, demonstrating its potential for reusable, cold-chain independent bioproduction and distribution channels. In doing so, this work presents to the best of our knowledge the first reported de novo production of various phenylpropenes from simple sugar starting materials as well as high ferulic acid de novo titers. This effort was demonstrated by successful accumulation of eugenol, the most chemically complex phenylpropene studied here. The modular construction allows for a unique bypassing of side products and enzyme promiscuities. However, enzyme promiscuities can likewise by leveraged for production of alternative products including hydroxychavicol and chavicol. Looking forward, we envision this modular approach can be expanded to additional plant secondary metabolite pathways currently hindered by enzymatic promiscuity[30]. While empirical evidence is often necessary to fully confirm enzymatic steps with the highest degree of substrate promiscuity in a pathway, computational tools such as SimZyme[47] as well as machine learning models[48,49], could be utilized in the absence of robust empirical evidence to gauge promiscuity and help predict best points in a pathway to split up into distinct modules and expedite co-culture prototyping.

## Methods

### Bacterial strains and plasmids

All plasmids/primers, and strains used in this study are listed in Supplementary Tables 1 and 2, respectively. *E. coli* DH10β was used for construction and amplification of plasmids. *E. coli* BL21(DE3) was used for feeding experiments and de novo production. Pathway construction was performed using plasmid backbones pETDuet-1, pACYCDuet-1, and pRSFDuet-1. Oligonucleotide primers used for PCR amplification were purchased from Integrated DNA Technologies (Coralville, IA). Strategies for individual plasmid construction are detailed in Supplementary Table 1. All heterologous genes were codon optimized for expression in *E. coli* and synthesized by Integrated DNA Technologies (Coralville, IA) or Twist Bioscience (San Francisco, CA). Plasmid pRSFDuet-1-fjTAL was a gift from Dr. Kristala L. J. Prather. The knockout strains were constructed by λ-RED recombination following standard protocols[50]. All strains listed in Supplementary Table 2, with the exception of the BL21(DE3) base strain, were transformed with listed plasmids via electroporation (2 mm Electroporation Cuvettes, BioExpress) with a BioRad Genepulser Xcell at 2.5 kV. Transformants were selected on Lysogeny broth agar plates containing appropriate antibiotics. All plasmid sequences in transformed strains were verified via Sanger Sequencing.

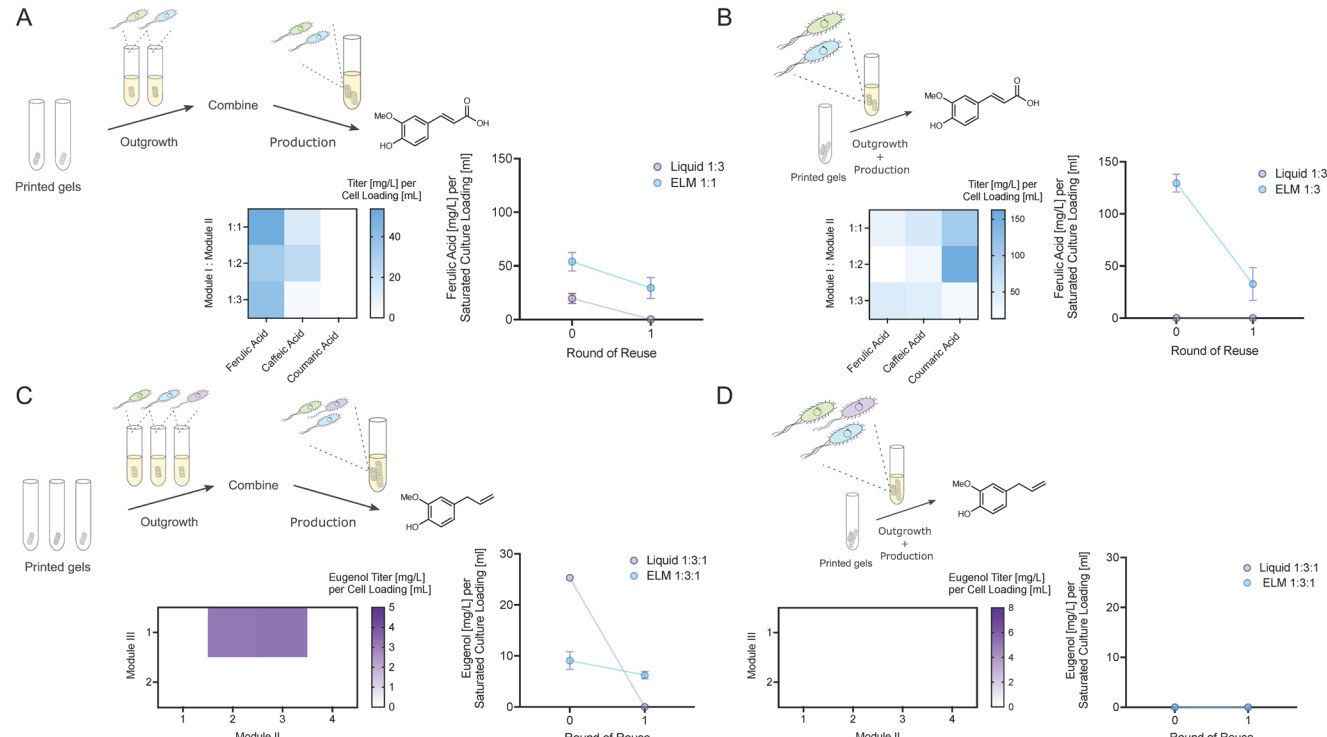

**Fig. 7 | Extension of biocatalytic efficiency and reusability through Engineered Living Material (ELM)-based production formats. A** Ferulic acid production using separate outgrowth and production stages. Heat map showcasing Round 0 ELM production titers of ferulic acid and hydroxycinnamic acid intermediates over range of Module I: Module II ratios. Line graph demonstrating head-to-head comparison of normalized titers for liquid and ELM constructs. **B** Ferulic acid production using combined outgrowth and production stages. Heat map showcasing Round 0 ELM production titers of ferulic acid and hydroxycinnamic acid intermediates over range of Module I: Module II ratios. Line graph demonstrating head-to-head comparison of normalized titers for liquid and ELM constructs. **C** Eugenol production using separate outgrowth and production stages. Heat map showcasing Round 0 ELM production titers of eugenol over range of Module II: Module III ratios. Line graph demonstrating head-to-head comparison of normalized titers for liquid and ELM constructs. **D** Eugenol production using combined outgrowth and production stages which showcases essentially no production in any condition. Heat map showcasing Round 0 ELM production titers of eugenol over range of Module II: Module III ratios. Line graph demonstrating head-to-head comparison of normalized titers for liquid and ELM constructs. Data are mean ± SD.; $n = 3$ biological replicates. Source data are provided as a Source Data file.

## Culture media and conditions

Strains were routinely cultured in Lysogeny broth (LB) medium containing yeast extract (5 g L$^{-1}$), tryptone (10 g L$^{-1}$), and NaCl (10 g L$^{-1}$) for inoculant preparation and cell propagation with appropriate antibiotic supplementation as needed (50 μg/ml Kanamycin and/or 30 μg/ml Chloramphenicol and/or 100 μg/ml Ampicillin). Modified MOPS medium was prepared for production experiments via supplementation of MOPS minimal medium[51] with thiamine (340 mg/L), glycerol (1%), glucose (0.25%), BME 100x vitamins (Millipore Sigma, 100x dilution), methionine (1 mM), Fe$_2$SO$_4$ (360 mg/L), potassium gluconate (1 mM), pantothenic acid (9.8 mg/L), niacin (112 mg/L), riboflavin (7.4 mg/L), IPTG (0.1 mM), and appropriate antibiotics (50 μg/ml Kanamycin and/ or 30 μg/ml Chloramphenicol and/or 100 μg/ml Ampicillin).

For deep well plate experiments, single colonies were inoculated into individual wells in 1 ml of LB media with appropriate antibiotics overnight at 37 °C. Overnight cultures were then diluted 50x into fresh LB and grown for 2 h at 37 °C to reach optical density at 600 nm (OD$_{600}$) ~ 1.0, confirmed via measurement of optical density at 600 nm using a spectrophotometer. Of note, all strains used in consortia tests in this study took approximately the same time to reach OD$_{600}$ ~ 1.0 prior to induction (Supplementary Table 3). IPTG was added to a final concentration of 0.1 mM and plates were grown for 5 h at 26 °C. Cells were then combined into desired co-culture ratios on a volume basis of induced culture, spun down for 5 min at 2000 × $g$ in a tabletop centrifuge, and resuspended in modified MOPS media for 63 h at 30 °C. For ferulic acid production, co-culture ratios were normalized such that '1' indicated a saturated culture (e.g. for deep well culture 1:3:1

indicated cells from 1 ml of Module I, 3 ml of Module II, and 1 ml of Module III were combined in the final production media). For eugenol production, co-culture ratios were normalized such that the total cells used corresponded to one saturated culture (e.g. for deep well culture 1:3:1 indicated cells from 200 μl of Module I, 600 μl of Module II, and 200 μl of Module III were combined in the final production media). For deep well feeding experiments, the same timeframes and dilution factors were utilized pre-production, but only 1 ml of a single module was resuspended in production media plus 2 mM of appropriate substrate (ferulic acid or coniferyl alcohol) for 48 h at 30 °C prior to analysis. While pre-induction OD$_{600}$ values were used to determine strain ratios during experimental setup, the colony-PCR quantified strain ratios post-induction and post-fermentation reflect differences between the modules when induced by IPTG and when in mixed culture settings (Supplementary Table 3). For co-culture experiments using a combined outgrowth and production scheme, IPTG was added when the full consortia was at OD$_{600}$ ~ 1.0. After the 5 h, 26 °C incubation step, the full consortia was again resuspended in production media for the production stage. For culture tube and flask experiments, all the same dilution factors and timing was used, with the volumes normalized to 2 ml and 30 ml final volumes, respectively. For deep well experiments in which rounds of reuse were utilized, liquid co-cultures were reused simply by sub culturing 20 μl of culture from the prior round into 1 ml fresh production media. Supernatant samples were taken at regular intervals for analysis of product formation via HPLC or LC-MS. Samples analyzed via HPLC were mixed with equal volumes of ethanol and filtered through a 0.22 μm membrane prior to

analysis. Samples analyzed via LC-MS were extracted into an equal volume of ethyl acetate and then filtered through a 0.22 μm membrane prior to analysis.

### ELM-based production experiments

*E. coli*-laden F127-BUM ELMs were prepared as previously described[31], with individual modules encapsulated into separate, 50 mg gels. Briefly, a 30 wt% F127-BUM polymer solution was mixed together with the photo-radical initiator 2-hydroxy-2-methylpropiophenone (Sigma-Aldrich; 2.5 μL was used for every 1 g of prepared hydrogel solution) to facilitate polymerization of the methacrylate functional groups upon subsequent UV exposure. Photo-initiated polymer solution was mixed with $4.5 \times 10^7$ cells prior to extrusion to yield robust, viable microbe-laden gels upon a brief photocuring step (achieved with an exposure to 365 nm light at $0.55\,mW\,cm^{-2}$ for a duration of 5 min using an Spectroline® XX-15NF model).

ELMs were cultured individually in wells of a 96 deep well plate, as described for the liquid cultures. ELMs culturing entailed the same timing for media exchange as for liquid cultures. For the case of monitoring growth in ELMs, cell density quantification within the gels has proven challenging given the rigid nature of the F127-BUM matrix[31]. To achieve a proxy for $OD_{600} \sim 1.0$ prior to induction with IPTG, ELMs were grown overnight to achieve saturation within the gel matrix. ELMS were then rinsed with fresh LB the following day, with timing for IPTG addition and transition to production media the same as described above for liquid cultures. The use of ELMs of equivalent size (50 mg) facilitated control over final consortia ratios, which could be achieved by adding specific numbers of ELMs for each strain. As cell sub culturing from ELMs is not possible, reuse was achieved via transferring ELMs to new media after a 2× rinse with sterile PBS. For the case of ELMs outgrown separately, ELMs were cultured in separate wells into the production media stage, in a similar manner as utilized in liquid cultures. For the case of ELMs grown together, ELMs were cultured in the same well after initial fabrication and kept within the same well for the duration of the experiment. Supernatant samples were taken at regular intervals for analysis of product formation via HPLC or LC-MS.

### HPLC analysis of products and intermediates

Products and pathway intermediates were analyzed by a Dionex Ultimate 3000 HPLC (Thermo Fisher Scientific) equipped with an Agilent Eclipse Plus C18 column (3.0 × 150 mm, 3.5 μm) with the UV detection wavelength set to 280 nm. The column oven was held at 25 °C with 1% acetic acid in water or acetonitrile as the mobile phase for a 20 min sequence subject to the following conditions: 5–15% organic (vol/vol) for 5 min, 15 to 100% organic (vol/vol) for 8 min, 100% organic (vol/vol) for 2 min, 100 to 5% organic for 2 min followed by 5% organic for 3 min. The flow rate was held constant at $0.8\,mL\,min^{-1}$. Coumaric acid, caffeic acid, ferulic acid, eugenol, coniferyl alcohol, and hydroxychavicol were quantified via comparison to standard curves made from serial dilutions of analytical standards. All standards were purchased from Millipore Sigma with the exception of hydroxychavicol, which was purchased from Fisher Scientific.

### LC–MS analysis of products and intermediates

For LC–MS analysis, the samples were analyzed using an Agilent Technologies 6125B Single Quadrupole LC-MS coupled with an ESI (electron spray ionization) source and interfaced with an Agilent 1200 series liquid chromatography system with a diode-array (UV–Vis) detector. The MS had an identification m/z range of 50–1500, with spectra acquired in positive mode for eugenol and chavicol and in negative mode for hydroxychavicol. The system was equipped with an Agilent ZORBAX Eclipse Plus C18 narrow bore column (2.1 mm internal diameter; 50 mm length; 5 micron particle size; P.N. 959746-902). The

LC method used a 12 min gradient ramp from 0.1% formic acid in water to 0.1% formic acid in acetonitrile with an additional 6 minute reset and equilibration time. The injection volume was 10 μL.

### Statistics and reproducibility

No statistical method was used to predetermine sample size. No data were excluded from the analyses. The experiments were not randomized. The investigators were not blinded to allocation during experiments and outcome assessment.

### Reporting summary

Further information on research design is available in the Nature Portfolio Reporting Summary linked to this article.

## Data availability

Data that support the findings of this study are presented in the main article and Supplementary Information files. Source data are provided with this paper. The plasmids and strains used are available from the corresponding author upon completion of a Material Transfer Agreement for academic use. Source data are provided with this paper.

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

## Acknowledgements

This work was supported by the National Science Foundation (NSF) Emerging Frontiers in Research and Innovation (EFRI) program under Award No. EFMA-2029249, received by H.S.A. S.M.B. acknowledges additional support from the NSF Graduate Research Fellowship Program. LC-MS data were collected with assistance from Ian Riddington at The University of Texas at Austin Department of Chemistry Mass Spectrometry Facility (MSF).

## Author contributions

S.M.B. and H.S.A. conceived the study. S.M.B designed and performed the experiments with help from C.M., K.B.R., S.-F.Y., DD.N., A.T., N.B., and G.A.-Y. C.M. performed LC–MS analysis of phenylpropenes. S.M.B., H.S.A., and A.N. wrote the manuscript.

## Competing interests

The authors declare no competing interests.
