## [Peer review file · Nature Communications]

REVIEWER COMMENTS

Reviewer #1 (Remarks to the Author):

In the article “A tripartite microbial co-culture system for de novo biosynthesis of diverse plant phenylpropanoids”, Brooks and co-workers reported a co-culture strategy using three engineered *E. coli* strains to produce plant-derived phenylpropanoids including eugenol, hydroxychavicol, and chavicol from sugar. This study proved the advantages of the co-culture system in metabolic engineering, especially for pathways with up to eight genes with promiscuity and toxic intermediates. Furthermore, the loading of the co-culture system on the living material strengthened its industrial potential as an alternative way to manufacture these compounds.

Overall, this is a well-written article with solid results. It demonstrates the optimization of phenylpropanoid production by splitting the heterologous pathway from tyrosine to eugenol into three different strains (at p-coumaric acid and ferulic acid, respectively). This strategy has been proven successful in some prior *E. coli* co-culture works. Representative examples include (a) a two-strain system converting glucose to p-coumaric acid and then to curcuminoid (Z. Fang, et al., *Biotechnol J*, 13, 2018), (b) a four-strain, 15-gene system converting sugars to anthocyanin (J.A. Jones, et al., *mBio*, 8, 2017), and (c) a three-strain, nine-gene system converting sugar to caffeic acid, salvianic acid A, and then rosmarinic acid (Z. Li, X. Wang, H. Zhang, *Metab Eng*, 54, 2019). While this work demonstrates the application of the co-culture strategy to the new and valuable phenylpropene biosynthesis, the paper might not be of general interest to the broader audience of *Nature Comm*. Another journal with a more specific group of audience might be proper.

A few other comments are listed below:

1. In Figure 2c, there is no negative control group (for example, WT *E. coli*) in the chromatography. Notably, a peak (RT~11.1) near the targeted peak in the “Consolidated” group is also shown later in the third group in Figure 6A. If it is the “non-phenylpropene side product” mentioned in Line 128, it would be helpful for the authors to characterize it with MS/MS or other analytical methods.
2. In Figure 3c, the third group of columns showed the titer when Modules I & II were co-cultured in media with yeast extract, Fe²⁺, and other vitamins. However, as Line 145 stated, "In an effort to remove the complexity of yeast extract in media, " yeast extract's involvement in this test needs more clarification.
3. Line 193 stated “high accumulation of chavicol,” but the titers of hydroxychavicol and chavicol produced were not reported except for the MS result.
4. Using ELM and the consequent enhancement is exciting, but more details would be helpful. For example, how to determine the mid-log phase for a three-module combination. It is also interesting to know whether the growth time for three kinds of cells to reach the mid-log phase is the same or different since they may have different growth burdens. How liquid and ELM were reused would also be included.

5. In the Introduction, many abbreviations were used directly without their full names shown in the first place.

6. In Figure 7a, only two modules were used, but the legend still shows “liquid 1:3:1”.

Reviewer #2 (Remarks to the Author):

In this manuscript, the authors designed a tripartite microbial co-culture system to enable the first de novo production of eugenol from simple sugar-starting materials. On this basis, they leveraged downstream enzyme promiscuity and modular design to achieve the production of hydroxychavicol and chavicol from simple sugar feed sources. They have further demonstrated the system’s compatibility with Engineered Living Material-based production which extends the biocatalyst efficiency and reusability of this consortia. The overall research is interesting. There are major and minor points the authors need to address.

Major points:

1. The authors wrote “a high number of heterologous enzymatic steps, pathway promiscuity, and microbial toxicity of pathway intermediates hinder efforts to effectively engineer a singular microbial strain from robust phenylpropenes synthesis.” The relevant references need to be cited to substantiate this claim. The authors specifically emphasize enzyme promiscuity throughout the entire text. Can authors provide an enzyme promiscuity definition? Is there a way to gauge the level of promiscuity so that one can decide whether to create consortia rather than a single strain for a particular biosynthesis?

2. The advantages in adapting the current system to ELMs are unclear. The authors should quantify the stability of the free culture consortia, since the stability of the consortia may not be a significant problem when the same species are used. The authors also said that the ELMs may be reused, but the results only indicated one round of recycling.

3. The author created a single strain in Fig. 2c with 8 exogenous enzymes involved in the manufacture of eugenol, but only discovered non-phenylpropene side products. How did the authors arrive at the conclusion that the fermentation product was not a phenylpropene side chemical and how did they test what the component was?

4. In Fig. 3d, the author adjusted the initial inoculation ratio of module 1 and module 2 to significantly improve the yield. Has the ratio of module 1 and module 2 been quantified during and after fermentation (same question with Fig. 4b)?

5. There is no description of SMB151 and SMB 146 in the main text. What is each substrate's conversion rate in the feeding experiment shown in Fig. 4a? Ferulic acid does not accumulate in the fermentation products, as shown in Fig. 4c, but the conversion rate of ferulic acid, which may be estimated from the

eugenol fermentation yield, is less than 20%. How different the outcome was from the feeding trial, and does other metabolic flux utilize ferulic acid?

6. The quantitative data in terms of yield and conversion rate should be shown in Figure 6 by the authors to examine the differences in hydroxychavicol yield between the tripartite co-culture and the tripartite co-culture [-COMT].

7. According to Fig. 7a and b, the accumulation of coumaric acid and caffeic acid differs depending on whether the outgrowth and production stages are separated or mixed. Can the authors explain these phenomena? Why does combined outgrowth produce more ferulic acid than separate outgrowth?

8. In Fig. 7c-d, results showed that no product was detected when all three strains were encapsulated in the same hydrogel. The authors claimed that the need for this separate outgrowth is likely due to the high promiscuity of the Module III enzymes for products formed by Modules I and II as well as intermediate product toxicities. This explanation is unclear. Can authors elaborate on these points since these three strains were indeed separating the pathway from each other (Figure 2).

9. Is the system scalable since most of the experiments were done below 100 mL.

Minor points

10. In line 225, should "(Fig. 3b)" be "(Fig. 3d)"?

11. In line 523, should "Module II (SMB150)" be "Module III (SMB150)"?

12. In fig. 7a, should "Liquid 1:3:1" be "Liquid 1:3"?

13. The format of the references needs to be modified and consistent.

14. There are numerous mistakes in the manuscript (line 260, 291 etc.).

15. The methods section is sparse - it lacks detail to evaluate, understand, and repeat the experimental design, and should be revised.

Response to Reviewer Comments

Reviewer #1 (Remarks to the Author):

In the article “A tripartite microbial co-culture system for de novo biosynthesis of diverse plant phenylpropanoids”, Brooks and co-workers reported a co-culture strategy using three engineered *E. coli* strains to produce plant-derived phenylpropanoids including eugenol, hydroxychavicol, and chavicol from sugar. This study proved the advantages of the co-culture system in metabolic engineering, especially for pathways with up to eight genes with promiscuity and toxic intermediates. Furthermore, the loading of the co-culture system on the living material strengthened its industrial potential as an alternative way to manufacture these compounds. Overall, this is a well-written article with solid results. It demonstrates the optimization of phenylpropanoid production by splitting the heterologous pathway from tyrosine to eugenol into three different strains (at p-coumaric acid and ferulic acid, respectively). This strategy has been proven successful in some prior *E. coli* co-culture works. Representative examples include (a) a two-strain system converting glucose to p-coumaric acid and then to curcuminoid (Z. Fang, et al., *Biotechnol J*, 13, 2018), (b) a four-strain, 15-gene system converting sugars to anthocyanin (J.A. Jones, et al., *mBio*, 8, 2017), and (c) a three-strain, nine-gene system converting sugar to caffeic acid, salvianic acid A, and then rosmarinic acid (Z. Li, X. Wang, H. Zhang, *Metab Eng*, 54, 2019). While this work demonstrates the application of the co-culture strategy to the new and valuable phenylpropene biosynthesis, the paper might not be of general interest to the broader audience of *Nature Comm*. Another journal with a more specific group of audience might be proper.

We thank the reviewer for this overview and comment. We have added the above references into the introduction section of the main text. Beyond more specialized journals, multiple studies detailing application of metabolic engineering tools to achieve *de novo* synthesis of industrially and/or pharmaceutically valuable products have been published in *Nature Communications* in recent years (e.g. W. Li, et al., *Nat Commun*. 13, 2022; Q. Liu, et al., *Nat. Commun*. 12, 2021; Y. Xu, et al., *Nat. Commun*. 13, 2022). We thank the reviewer for their overall comments regarding this work to be new and valuable. We respectfully disagree with the choice of journal. Given the potential industrial interest of our target phenylpropene products, demonstrated utility of ELM-based manufacturing, as well as demonstration of the use of promiscuity to leverage the same base consortia to make multiple distinct and valuable phenylpropene products, we feel the broader audience of *Nature Communications* to be a proper journal for this work.

A few other comments are listed below:

1. In Figure 2c, there is no negative control group (for example, WT *E. coli*) in the chromatography. Notably, a peak (RT~11.1) near the targeted peak in the “Consolidated” group is also shown later in the third group in Figure 6A. If it is the “non-phenylpropene side product” mentioned in Line 128, it would be helpful for the authors to characterize it with MS/MS or other analytical methods.

We thank the reviewer for this comment. We have added the control HPLC trace for the control group (BL21(DE3) WT *E. coli*). Regarding the RT ~11.1 peak, this was confirmed by LC-MS to

not contain any of the three target phenylpropenes of this work (eugenol, hydroxychavicol, or chavicol). The peak was also produced by the coumaric acid producer alone, as well as the coculture [-C3H -COMT] (now reflected in Supplemental Figure 1) and thus we believe it is likely an upstream metabolite related to tyrosine and/or coumaric acid biosynthesis. This has been reflected in the text. Unfortunately, the compound does not ionize well on LC-MS or GC-MS and we were unable to obtain a predicted mass.

2. In Figure 3c, the third group of columns showed the titer when Modules I & II were co-cultured in media with yeast extract, Fe²⁺, and other vitamins. However, as Line 145 stated, "In an effort to remove the complexity of yeast extract in media, " yeast extract's involvement in this test needs more clarification.

We thank the reviewer for this comment. While yeast extract is indeed used in the third group of columns, in the text we called out the formulation which enabled the highest ferulic acid titer (the fourth column) which eliminates yeast extract as an ingredient. The involvement of yeast extract in this experiment, as well as the lack of a need for it in the final formulation to achieve the highest *de novo* ferulic acid titer, has been clarified in the text.

3. Line 193 stated "high accumulation of chavicol," but the titers of hydroxychavicol and chavicol produced were not reported except for the MS result.

We thank the reviewer for this comment. We have quantified the titers of hydroxychavicol via an analytical standard and have included the titers of hydroxychavicol produced from the tripartite coculture and tripartite coculture [-COMT] in the main text. We were unable to obtain an analytical standard for chavicol so still remain unable to quantify it. The initial statement about high accumulation was primarily referencing relative peak areas between the chavicol peak in the coculture [-COMT -C3H] and eugenol peak in the full tripartite coculture. We have revised to text to state 'accumulation of chavicol' to avoid any confusion.

4. Using ELM and the consequent enhancement is exciting, but more details would be helpful. For example, how to determine the mid-log phase for a three-module combination. It is also interesting to know whether the growth time for three kinds of cells to reach the mid-log phase is the same or different since they may have different growth burdens. How liquid and ELM were reused would also be included.

We thank the reviewer for this comment. We have added more details to the methods section regarding liquid and ELM culture and production setups to make the processes clearer. In the case of the liquid three-module combination, the growth phase was determined for the entire consortia via optical density measurements at 600nm, in which production was initiated when the full consortia was at OD₆₀₀ = 1.0. For ELM constructs, determination of cell density within the gels has proven challenging given the rigid nature of the F127-BUM matrix. We have discussed this phenomenon in prior works (e.g. Johnston *et al.* Nat Comm. 2020). To achieve a proxy for log phase prior to induction with IPTG, ELMs were grown overnight to achieve saturation within the gel matrix. ELMs were then rinsed with fresh LB the following day, with timing for IPTG addition and transition to production media the same as described above for liquid cultures. This

procedure was the same regardless of the number of modules combined at once and has been clarified in the methods section. Regarding the time to reach log phase, we have observed that all strains used in this study take approximately the same amount of time to reach an $OD_{600} = 1.0$ and have stated this in the text. Thus, we expect the growth burdens to be similar. The procedures for liquid and ELM reuse have been clarified in the methods section.

5. In the Introduction, many abbreviations were used directly without their full names shown in the first place.

These have been resolved in the text.

6. In Figure 7a, only two modules were used, but the legend still shows “liquid 1:3:1”.

This has been resolved in the figure.

Reviewer #2 (Remarks to the Author):

In this manuscript, the authors designed a tripartite microbial co-culture system to enable the first de novo production of eugenol from simple sugar-starting materials. On this basis, they leveraged downstream enzyme promiscuity and modular design to achieve the production of hydroxychavicol and chavicol from simple sugar feed sources. They have further demonstrated the system’s compatibility with Engineered Living Material-based production which extends the biocatalyst efficiency and reusability of this consortia. The overall research is interesting. There are major and minor points the authors need to address.

Major points:

1. The authors wrote “a high number of heterologous enzymatic steps, pathway promiscuity, and microbial toxicity of pathway intermediates hinder efforts to effectively engineer a singular microbial strain from robust phenylpropenes synthesis.” The relevant references need to be cited to substantiate this claim. The authors specifically emphasize enzyme promiscuity throughout the entire text. Can authors provide an enzyme promiscuity definition? Is there a way to gauge the level of promiscuity so that one can decide whether to create consortia rather than a single strain for a particular biosynthesis?

We thank the reviewer for this comment. We have added the relevant references to substantiate this sentence into the text, with additional references in the results section to further support promiscuity and toxicity aspects of the pathway. Additionally, we have added an enzyme promiscuity definition to the introduction of the main text to clarify this meaning upfront. Specifically, we add “For clarity, this text defines promiscuity within the phenylpropene pathway as the ability of multiple pathway enzymes to perform the same type of reaction on multiple substrates (referred to as substrate promiscuity in other contexts³⁰). For instance, the set of enzymes responsible for conversion of ferulic acid to coniferyl alcohol have also been shown to

readily act on available pools of coumaric acid and caffeic acid in prior works²⁸”. The numerical references refer to numbered references found in the ‘References’ section of the main text.

In terms of gauging promiscuity, it is well documented that several plant secondary metabolite pathways of industrial and/or pharmaceutical interest, including phenylpropanoid, flavonoid, and alkaloid biosynthesis, have a high incidence of substrate promiscuity (T. Waki, et. al., *BioEssays*, 43, 2021). It is believed that ~40% of enzymes have promiscuous function (H. Nam, et al., *Science*, 337, 2012). This problem comes to light more significantly in engineered cells where substrate concentrations can be multi-fold higher than what is found under natural evolution conditions. Thus, we envision this co-culture strategy to be relevant to the majority of microbial systems aimed at complex plant secondary metabolite biosynthesis and broadly across other metabolic pathways.

Though empirical evidence is often necessary to fully confirm an enzyme or pathway’s promiscuity, there are several documented methods to gauge enzymatic promiscuity computationally. For example, substrate similarity-based search tools, such as SimZyme, (D. Pertusi, et al., *Bioinformatics*, 31, 2015) use 2D chemical fingerprints to search for promiscuous substrates an enzyme may act on within a metabolic network. Of particular recent interest are machine-learning models (G. M. Visani, et al., *Bioinformatics*, 37, 2021 and S. Goldman, et al., *PLoS Comput Biol*, 18, 2022) to predict enzyme promiscuity. This concept has been added to the Discussion in the main text.

2. The advantages in adapting the current system to ELMs are unclear. The authors should quantify the stability of the free culture consortia, since the stability of the consortia may not be a significant problem when the same species are used. The authors also said that the ELMs may be reused, but the results only indicated one round of recycling.

We thank the reviewer for this comment. We have verified the lack of stability in both the ferulic acid co-culture and full eugenol tripartite co-culture via the observation of lack of production of the target compounds from the liquid culture mode after a single round of reuse, as shown in Figure 7. This was quantified both with respect to product levels (Figure 7) and the quantification of strain ratios at the beginning and end of fermentation (Supplemental Table 3). This has been highlighted in the text. While only a single round of reuse was tested in this work as a proof-of-concept, we anticipate that multiple additional rounds of reusability could be attainable as shown in prior works showcasing reusability of F127-BUM ELMs (T. Johnston, et. al., *Nat. Commun.*, 11, 2020). This has also been highlighted in the text.

3. The author created a single strain in Fig. 2c with 8 exogenous enzymes involved in the manufacture of eugenol, but only discovered non-phenylpropene side products. How did the authors arrive at the conclusion that the fermentation product was not a phenylpropene side chemical and how did they test what the component was?

We thank the reviewer for this comment. This side product produced by this strain was also observed to be produced by the coumaric acid module alone, now shown in Supplemental Figure 1, which lacks any enzymes for producing phenylpropenes. When analyzed via LC-MS, the peak

does not contain any of the masses of the three target phenylpropenes of this work (eugenol, hydroxychavicol, or chavicol) and is at a different retention time than any of the target phenylpropenes, as evident by the LC spectra shown in the Supplementary Information. Given that the peak was produced by the coumaric acid producer alone, as well as the coculture [-C3H - COMT] (now also reflected in Supplemental Figure 1), we believe it is likely an upstream metabolite related to tyrosine and/or coumaric acid biosynthesis. This has been reflected in the text. Unfortunately, the compound does not ionize well on LC-MS or GC-MS and we were unable to obtain a predicted mass.

4. In Fig. 3d, the author adjusted the initial inoculation ratio of module 1 and module 2 to significantly improve the yield. Has the ratio of module 1 and module 2 been quantified during and after fermentation (same question with Fig. 4b)?

We thank the reviewer for this comment. For all module ratio testing experiments, the ratios were based on OD₆₀₀ values of each strain prior to IPTG induction, with desired ratios achieved by adding different volumes of starting culture from each module into the combined production stage, as described in the methods section. We have additionally quantified ratios of Module I and II, as well as Module I, II, and III, both post-induction and post-fermentation and included this data as Supplemental Table 3.

5. There is no description of SMB151 and SMB 146 in the main text. What is each substrate's conversion rate in the feeding experiment shown in Fig. 4a? Ferulic acid does not accumulate in the fermentation products, as shown in Fig. 4c, but the conversion rate of ferulic acid, which may be estimated from the eugenol fermentation yield, is less than 20%. How different the outcome was from the feeding trial, and does other metabolic flux utilize ferulic acid?

We thank the reviewer for this comment. We have added a brief description of the function of SMB151 and SMB146 in the main text and Fig. 4a, with a full description in Supplementary Table 2. The conversion rate of each strain for its substrate has been added to Fig. 4a, and ranges from ~70% - 80% for each strain. The conversion rate for the eugenol tripartite consortium, estimated from the maximum stoichiometric ferulic acid titer is about 22% using a molar basis. However, as noted in the methods, the eugenol consortia with maximum production utilized a 4x lower cell density than that of ferulic acid. Thus, the conversion using a per cell basis was comparable to the feeding assays. Beyond this result, we are unsure exactly why higher cell density does not then further increase eugenol titer (as shown in Supp. Fig. 3). It is possible that the metabolic flux of the coumaric acid and ferulic acid modules changes in the presence of the phenylpropene module and limits max ferulic acid available for conversion during the tripartite coculture. Future work investigating this phenomenon could be very valuable to the field and help elucidate how to further boost phenylpropene titers.

6. The quantitative data in terms of yield and conversion rate should be shown in Figure 6 by the authors to examine the differences in hydroxychavicol yield between the tripartite co-culture and the tripartite co-culture [-COMT].

We have quantified the relative titers of hydroxychavicol, as well as the final molar ratio of eugenol to hydroxychavicol, for the tripartite coculture and tripartite coculture [-COMT] and added the metrics to the Supplementary Information as Supplemental Table 4.

7. According to Fig. 7a and b, the accumulation of coumaric acid and caffeic acid differs depending on whether the outgrowth and production stages are separated or mixed. Can the authors explain these phenomena? Why does combined outgrowth produce more ferulic acid than separate outgrowth?

We thank the reviewer for this comment. We agree that the great difference in accumulation of precursors coumaric acid and caffeic acid, as well as final ferulic acid titer, between culture modes is quite interesting. At this time, we are unsure of the mechanism underlying the difference in metabolic fluxes and final product profiles between culture modes. This could be a great topic for in-depth future study, and we hope that by displaying the results and pointing out the large difference that we can inspire others to help uncover this mechanism in future works.

8. In Fig. 7c-d, results showed that no product was detected when all three strains were encapsulated in the same hydrogel. The authors claimed that the need for this separate outgrowth is likely due to the high promiscuity of the Module III enzymes for products formed by Modules I and II as well as intermediate product toxicities. This explanation is unclear. Can authors elaborate on these points since these three strains were indeed separating the pathway from each other (Figure 2).

We thank the reviewer for this comment. We are not certain of the exact mechanism underlying this phenomenon but have provided a more complete proposed mechanism in the text. Specifically, that “The need for this separate outgrowth stage is likely due to the consortia instability and depletion of Module I during stages in which all modules are grown together (Supplementary Table 3). This instability could arise from a variety of mechanisms including intermediate product toxicities^{4,13,29} or a growth advantage of Module II, potentially leading to depletion of Module I over time. Further investigation into this phenomenon is warranted in future studies to elucidate the exact impact of combined vs separate outgrowth on ELM production.” The numerical references refer to numbered references found in the ‘References’ section of the main text.

9. Is the system scalable since most of the experiments were done below 100 mL.

We thank the reviewer for this comment. As shown in the text, we demonstrated eugenol production at both 1 ml and 30 ml scales, in which similar titers were observed. Thus, we feel confident that further scale-up should enable at least the same titers, if not higher, with additional pH and carbon source feeding strategy optimization. Future works could investigate bioreactor scale of phenylpropenes to further probe industrial scale-up potential, but this was not the main focus of our study.

Minor points

10. In line 225, should “(Fig. 3b)” be “(Fig. 3d)”?

This has been resolved in the text

11. In line 523, should “Module II(SMB150)” be “Module III(SMB150)”?

This has been resolved in the text.

12. In fig. 7a, should “Liquid 1:3:1” be “Liquid 1:3”?

This has been resolved in the figure.

13. The format of the references needs to be modified and consistent.

This has been resolved in the text.

14. There are numerous mistakes in the manuscript (line 260, 291 etc.).

This has been resolved in the text.

15. The methods section is sparse - it lacks detail to evaluate, understand, and repeat the experimental design, and should be revised.

This has been resolved in the text.

REVIEWERS' COMMENTS

Reviewer #2 (Remarks to the Author):

The authors have adequately addressed my concerns in the revision. Some other unsolved problems are also discussed by the authors and marked in the original text. I support the publication and suggest that the authors thoroughly checked the text (in Fig. 7a, should " ELM 1:1" be "ELM 1:3"?) and reference format.

REVIEWERS' COMMENTS

Reviewer #2 (Remarks to the Author):

The authors have adequately addressed my concerns in the revision. Some other unsolved problems are also discussed by the authors and marked in the original text. I support the publication and suggest that the authors thoroughly checked the text (in Fig. 7a, should “ELM 1:1” be “ELM 1:3”?) and reference format.

We thank the reviewer for this comment. We have thoroughly checked the text and reference format in this final version. In Fig. 7a the ELM 1:1 label is correct – as evident in the heat map in that figure the highest ferulic acid titer in the separate outgrowth format for ELM culture was achieved with the 1:1 ratio. Thus, we utilized that ratio in the graph. As stated in the text, the 1:3 ratio used for liquids did enable the highest selectively for ferulic acid buildup over caffeic acid or coumaric acid in the ELM format but the purpose of the graph in question was showcasing maximum ferulic acid titers achieved by each culture mode.